# Changes in Attitudes towards Influenza and Pneumococcal Vaccination during the Subsiding COVID-19 Pandemic—Results of a Longitudinal Survey Study among Risk Groups in Germany between 2021 and 2023

**DOI:** 10.3390/vaccines12091080

**Published:** 2024-09-21

**Authors:** Sebastian Born, Daniel Schwarzkopf, Norman Rose, Mathias W. Pletz, Konrad Reinhart, Carolin Fleischmann-Struzek

**Affiliations:** 1Institute of Infectious Diseases and Infection Control, Jena University Hospital, 07743 Jena, Germany; norman.rose@med.uni-jena.de (N.R.); mathias.pletz@med.uni-jena.de (M.W.P.); carolin.fleischmann-struzek@med.uni-jena.de (C.F.-S.); 2Center for Sepsis Control and Care, Jena University Hospital, 07743 Jena, Germany; 3Department of Anesthesiology and Intensive Care Medicine, Jena University Hospital, 07743 Jena, Germany; daniel.schwarzkopf@med.uni-jena.de; 4Department of Anesthesiology and Intensive Care Medicine, Charité—Universitätsmedizin Berlin, 13353 Berlin, Germany; konrad.reinhart@charite.de; 5Sepsis Foundation, 12203 Berlin, Germany

**Keywords:** vaccination attitude, influenza, pneumococcal, COVID-19, risk perception

## Abstract

Background: In many countries, an increase in influenza and pneumococcal vaccination rates was observed during the COVID-19 pandemic. We examined how attitude, risk perception and knowledge towards influenza and pneumococcal vaccines of at-risk patients developed when the COVID-19 pandemic subsided and if COVID-19 vaccination attitude (VA) was still associated with the attitudes towards the two other vaccines. Methods: We used longitudinal data from two surveys conducted in Germany in 2021 and 2023 among persons with chronic diseases. We assessed VA, risk perception, vaccination knowledge and further psychological determinants of vaccine acceptance. Structural equation modelling using full information maximum likelihood was used to estimate multivariate regressions with planned missing data. Results: Among 543 respondents, the influenza and pneumococcal vaccination rates remained relatively stable between 2021 and 2023. VA also remained unchanged at a moderately positive level, while COVID-19 VA decreased. A constantly positive association between COVID-19 VA and influenza as well as pneumococcal VA was found, independent from a general VA. The perceived danger of influenza increased between 2021 and 2023 and was among the strongest predictors of influenza VA. Conclusions: Also at the subsiding pandemic, COVID-19 VA was constantly associated with the influenza and pneumococcal VA. It seems sensible to take these aspects into account when designing future vaccination campaigns for at-risk patients. Trial registration: DRKS00024561. Registered 9 March 2021.

## 1. Introduction

Influenza and pneumococcal vaccination are effective measures to prevent life-threatening complications such as pneumonia and sepsis in the at-risk population [1,2]. Therefore, regular vaccinations are recommended for persons aged ≥ 60 years and those living with chronic immunosuppressive comorbidities by the WHO and many national immunization plans [3]. Although efforts were made to promote vaccination and inform at-risk populations on the benefits of the vaccination, uptake of both vaccinations persisted at a low level in many countries [3,4,5]. However, an increase in influenza and pneumococcal vaccination rates in adults was observed during the first and second waves of the COVID-19 pandemic after public authorities advocated these vaccines to avoid a “twindemic” [6,7]. Furthermore, the shortage of pneumococcal vaccines in 2020 [8] may have led to a rising demand. The contemporary increase in vaccination rates was interpreted as a spill-over effect from COVID-19, as was the finding that influenza was perceived as more dangerous, while vaccinations were evaluated as safer and more beneficial during the pandemic, according to a Finnish survey study [9]. Generally, a spill-over effect describes a change in attitude regarding a specific concept that spills over to related concepts [10]. In a meta-analysis based on 22 studies, an increase in influenza vaccination intention in the season 2020/2021 at the peak of the COVID-19 pandemic was found adjusting for age, gender or education [11]. That review also found explicitly that worries about COVID-19 increased influenza vaccine uptake in many countries [11]. In an US-wide ecological study, adult influenza vaccine uptake was decreased within states with low COVID-19 vaccine uptake, while an increase was found within states with high uptake [12]. For the pneumococcal vaccination, similar patterns in vaccination uptake were found during the pandemic in selected countries [13]. Two French survey studies conducted in 2020/2021 showed that the COVID-19 pandemic was named as an important contributor to the positive decision to get vaccinated against pneumococci [14], and that the pandemic convinced them to get both influenza and pneumococcal vaccinations [15]. 

While the evidence from these studies may indicate a positive spill-over from the COVID-19 pandemic and vaccination attitude into influenza and pneumococcal vaccination attitudes during the peak of the pandemic, little is known on pandemic-related influences at the subsiding of the pandemic. It remains unclear if the association between COVID-19 and influenza as well as pneumococcal vaccination attitudes continued even after the immediate threat of the pandemic had disappeared. Furthermore, we lack a deeper understanding of how other psychological determinants that are considered relevant for vaccination uptake, such as risk perception with regard to the disease and the vaccination, or vaccination knowledge [16], may have changed over time. 

In the present study, we therefore aimed to examine (1) how the vaccination rates, attitude, risk perception and knowledge towards influenza and pneumococcal vaccines of at-risk patients developed when the COVID-19 pandemic subsided; (2) if the attitude towards the COVID-19 vaccination is associated with the attitudes towards the two other vaccines; and (3) if these associations change over time.

## 2. Methods

### 2.1. Study Design and Setting

The longitudinal data were obtained by two sequential surveys between 22 March and 8 July 2021 and between 21 March and 4 July 2023 (preregistration: DRKS00024561). It is part of the SepsisWissen (SepWiss) intervention study (Strengthening the health literacy of patients at high risk for sepsis to improve early diagnosis and treatment of sepsis; preregistration: DRKS00024475), which aims to increase sepsis knowledge and the ability to recognize sepsis as an emergency by a multifaceted information campaign in the German states of Berlin and Brandenburg. Part of the evaluation involves a longitudinal survey among risk groups. The current study reports findings from the first and third waves of the risk group survey (pre- and post-intervention); however, the intervention was not found to have any significant effect on the vaccination attitude in the model region. Ethical approval for the study was obtained from the Institutional Review Board of the Friedrich-Schiller University Jena (2020-1921-Bef). The reporting of the study followed the checklist for reporting of survey studies (CROSS) [17].

### 2.2. Sample Recruitment and Survey Conduction

For the first wave of the longitudinal study, the market research institute IPSOS recruited a sample of 543 persons from Germany aged at least 18 years. In accordance with the study design, 260 of the initial 543 persons were surveyed again in the third wave. The recruitment process was designed to reach a predetermined proportion of patients with certain chronic diseases (e.g., diabetes, chronic diseases, cancer) in each wave. The proportions were the same across all waves. The sample was recruited via self-help groups, general practitioners, social media and contacts with participants from previous studies. Persons who gave their informed consent to participate in the study were interviewed online, by telephone or in person using a standardized questionnaire.

### 2.3. Survey Development

The initial aim of the survey was to evaluate the multimodal information campaign aimed at increasing the sepsis knowledge and the vaccination rates for influenza and pneumococci. In addition to the constructs relevant for the evaluation, the general and specific vaccination attitude and the risk perception of the three diseases (influenza/pneumococci/COVID-19) were recorded. We used validated questions and scales from previous studies [16,18,19,20]. For the development of a questionnaire, we followed recommended methods [21,22]: After selecting the relevant constructs based on theoretical considerations and the existing research literature, we created an item pool and drafted the first version of the survey. Based on the results of cognitive interviews with different experts in the fields of emergency medicine, infectious diseases, intensive care medicine and psychiatry and one patient (*n* = 6), we revised the survey and conducted a pretest among members of self-help groups for different chronic diseases (*n* = 71) as an online survey. The results of the pretest were used to develop the final version of the survey.

Throughout the paper, we will use the terms first and second time point for the first and third wave of the risk group survey.

### 2.4. Variables

We recorded age, sex, educational level, employment state and health insurance type as sociodemographic data. Furthermore, we assessed the comorbidities and the vaccination status with regard to influenza and pneumococcal vaccination via self-report.

#### 2.4.1. Vaccination Attitude

Attitude towards vaccination is one of the most relevant psychological determinants for vaccine hesitancy and vaccine uptake [16]. Following the study from Askelson et al. [18], vaccination attitude was assessed with three items for each vaccination (influenza/pneumococci/COVID-19) in our survey. We asked the participants whether vaccination against influenza/pneumococci/COVID-19 (I) “is necessary”, (II) “is a good idea” and (III) “is beneficial”. Agreement with the statements was recorded on a five-point Likert scale. Following Askelson et al., the mean of the three items served as the measure of attitude, with higher values indicating a more positive attitude towards the corresponding vaccination.

#### 2.4.2. Risk Perception

We assessed risk perception regarding influenza/pneumococci/COVID-19 with four items used in previous studies, covering the perceived danger of the disease, the perceived susceptibility for the disease, the perceived danger of the vaccination and the perceived effectiveness of the vaccination [20]. Participants rated the statements on a five-point Likert scale ranging from “very low” to “very high”. For COVID-19 we only used the perceived danger of the vaccination and the perceived effectiveness of the vaccination.

#### 2.4.3. Vaccination Knowledge

Vaccination knowledge was recorded using one item for each vaccination. The participants had to decide (correct/incorrect) whether vaccination against influenza or pneumococci is recommended for people of their age and with their comorbidities. Note that both the influenza and pneumococcal vaccination were indicated for all people in our sample.

#### 2.4.4. Psychological Antecedents of Vaccination (5C)

We used the 5C scale (confidence, calculation, constraints, complacency, collective responsibility) as a measure for general vaccination attitude in contrast to the vaccination-specific attitude (influenza, pneumococci, COVID-19). Participants had to rate 15 items (three items per antecedent) on a five-point Likert scale ranging from “totally disagree” to “totally agree”. The score for each of the psychological antecedents was calculated as mean. Negatively poled items were recoded before calculating the mean.

### 2.5. Analyses

Descriptive statistics were calculated for sociodemographic information and patient data; means and standard deviations (SDs) for metric data; frequencies and percentages for nominal and ordinal data. The reliability of the vaccination attitude scale and the subscales of the psychological antecedents (5C) were assessed by McDonald‘s Omega [23], which has proven superior to Cronbach’s α in cases of unequal factor loadings and missing item responses [24].

To examine potential predictors of vaccination attitude, we calculated two separate models for influenza and pneumococci. For each model, we estimated a multivariate regression with vaccination attitude at the first and second time point as the two outcomes using structural equation modelling (SEM). This approach allows for a direct comparison of means and regression coefficients for the two time points within the model. The set of predictors consists of risk perception regarding influenza and pneumococcal infections (e.g., danger of the disease, risk to get the disease, danger of the vaccination, effectiveness of the vaccination), vaccination knowledge of the corresponding vaccination, the vaccination attitude and risk perception regarding COVID-19 (danger of the vaccination, effectiveness of the vaccination), as well as the five scores of the 5C scale. Age, sex and educational level were only used as auxiliary variables to handle the missing data. We reported standardized partial regression coefficients β as measures of the conditional linear relationship between a single predictor and the outcome under statistical control of the remaining predictors in the model. To check for possible changes in the conditional stochastic associations between predictors and outcomes over time, we tested the differences in partial regression coefficients using the Delta method [25]. Additionally, we provide R^2^ values (e.g., proportion of explained variance) of each outcome as a summary measure of the predictive power of the regression model. We report unadjusted *p*-values with a significance level α = 0.05. 

Note that our data set contains planned missing data by design, as only a proportion of the original sample was interviewed again at the second time point [26,27]. We utilized full information maximum likelihood (FIML) estimation, which allows for inclusion of all participants including those with incomplete data due to non-participation at the second time point. This is a major advantage of using SEM for parameter estimation of multivariate linear regression models [28,29]. All analyses were performed using R Version 4.2.3 [30]. The R-package lavaan [31] was used for FIML of the multivariate regression model. 

## 3. Results

### 3.1. Survey Participants

Figure 1 shows the survey time periods in relation to relevant COVID-19 vaccination-related developments in Germany. A total of 543 individuals completed the survey at the first time point, of which 260 participated at the second time point (47.9%, Figure 1). Demographic features of survey participants are shown in Table 1.

### 3.2. Temporal Trends in Vaccination Rates, Vaccination Attitude, Risk Perception and Vaccination Knowledge

The vaccination rate in our survey population for influenza in 2021 (41.6%) was significantly higher than the vaccination rate for pneumococcal vaccination (16.0%). In 2023, the vaccination rates for influenza and pneumococcal vaccination remained almost unchanged (39.6% and 17.7%).

At the first time point, attitude towards influenza and pneumococcal vaccination was moderately positive (M = 3.384 and M = 3.116 out of 5 points Likert scale, respectively), but less positive than the attitude towards COVID-19 vaccination (M = 4.188). The attitude towards influenza and pneumococcal vaccination remained stable between the first and the second time point (Table 2), while the average positive attitude towards COVID-19 vaccination decreased significantly (M_2_ − M_1_ = −0.714, *p* < 0.001, Appendix A). For the vaccination attitude scales, McDonald’s Omega reliability estimate ranged from 0.91 to 0.96, which indicates a high reliability. For more details, see Appendix A.

Risk perception is represented by Danger Disease, Risk to Get, Danger Vaccination and Effectiveness Vaccination. Value range for the means of all variables except Vaccination Knowledge is 1 to 5. Values for the mean of Vaccination Knowledge range from 0 to 1 and correspond to the proportion of respondents knowing that vaccination against influenza or pneumococcus is recommended for people of their age and with their comorbidities.

With regard to risk perception, the danger of an influenza infection was perceived higher at the second survey (M_2_ − M_1_ = 0.294, *p* < 0.001), whereas the perceived danger of a pneumococcal infection remained unchanged. However, respondents rated the risk of becoming infected with influenza and pneumococci significantly higher at the second survey (M_2_ − M_1_ = 0.158, *p* < 0.001, and M_2_ − M_1_ = 0.193, *p* < 0.001, respectively). Contrary, the perceived effectiveness of both vaccines was rated lower (M_2_ − M_1_ = −0.183, *p* < 0.001, and M_2_ − M_1_ = −0.208, *p* < 0.001, respectively) at the second time point. The perceived danger of both vaccinations remained stable over time on a low level (Table 2).

In both surveys, vaccination knowledge was higher for the influenza vaccination than for the pneumococcal vaccination. About three out of four respondents knew that the influenza vaccination is recommended for people of their age and comorbidities. This proportion remained stable over time. Although the level of knowledge about pneumococcal vaccinations increased slightly from the first to the second survey (M_2_ − M_1_ = 0.047, *p* = 0.035), only around half of respondents knew that this vaccination was recommended for them.

### 3.3. Predictors of Vaccination Attitude

In the multivariate regression analyses for the year 2021 (first time point), a more positive attitude towards the influenza vaccination (Table 3) was associated with a higher perceived danger of influenza (β = 0.204), a higher perceived risk to contract influenza (β = 0.224), a higher perceived effectiveness of the influenza vaccination (β = 0.150), a better influenza vaccination knowledge (β = 0.125), a more positive COVID-19 vaccination attitude (β = 0.243) and a higher general confidence in vaccinations (β = 0.193). Furthermore, we found negative associations with the perceived danger of the influenza (β = −0.083) and the perceived effectiveness of the COVID-19 vaccination (β = −0.140). No associations were found with the danger of the COVID-19 vaccination and the remaining aspects of the 5C scale (calculation, constraints, complacency, and collective responsibility). The model explained R^2^ = 64.5% of the variance in influenza vaccination attitude.

In 2023 (second time point), only the perceived danger of influenza (β = 0.164), perceived risk to contract influenza (β = 0.337) and the COVID-19 vaccination attitude (β = 0.232) remained significant predictors for the attitude towards influenza vaccination. However, a significant change in the association (e.g., a difference in the partial regression coefficients over time) was only observed for the risk to get influenza. The model explained R^2^ = 68.0% of the variance in influenza vaccination attitude.

For 2021, a positive attitude towards pneumococcal vaccination (Table 4) was associated with a higher perceived danger of pneumococcal infections (β = 0.180), a higher perceived risk to contract pneumococcal infections (β = 0.210), a better pneumococcal vaccination knowledge (β = 0.230), a more positive COVID-19 vaccination attitude (β = 0.161) and a higher general confidence in vaccinations (β = 0.107). The aspects calculations (β = −0.042) and constraints (β = −0.109) of the 5C scale were negatively associated with pneumococcal vaccination attitude. The remaining predictors were not significantly associated with the attitude towards vaccination. The model with all predictors explained R^2^ = 56.7% of the variance in pneumococcal vaccination attitude.

Similar to influenza, only the perceived danger of pneumococcal infections (β = 0.218), the perceived risk to get a pneumococcal infection (β = 0.238) and the COVID-19 vaccination attitude (β = 0.258) remained significant predictors for the attitude towards pneumococcal vaccination in 2023. A significant change in the association was only observed for the pneumococcal vaccination knowledge. The model explained R^2^ = 64.3% of the variance in pneumococcal vaccination attitude. 

## 4. Discussion

Our results expand the existing body of literature with new insights on the relation between vaccination attitudes at the subsiding COVID-19 pandemic. We observed stable influenza and pneumococcal vaccination rates among risk patients, which however remained rather low and below the recommendations of the WHO (e.g., 75% for influenza in the age group ≥ 65 years [32]). Similarly, the moderately positive attitude towards the influenza and pneumococcal vaccination remained unchanged between 2021 and 2023, while the initially very positive COVID-19 vaccination attitude considerably decreased. Nonetheless, for both survey time points, we found a positive association between attitude towards COVID-19 vaccination and influenza as well as pneumococcal vaccination attitude, independent from a general vaccination attitude captured by the 5Cs scale [19]. Furthermore, perceived risk of contracting influenza or pneumococcal infection increased from 2021 to 2023, which we hypothesize may be associated with the removal of COVID-19 contact restrictions and other prevention strategies in 2023. Likewise, the perceived danger of influenza also increased over time, but such a trend was not observable for pneumococcal infections.

Our results differ from the recent literature in two main points: First, we did not find a decline in influenza and pneumococcal vaccination rates compared to the peak of the pandemic, which was observed by different previous studies from other countries, e.g., for France and the US [33]. Second, while previous studies interpreted increasing influenza and pneumococcal vaccination rates and more positive vaccination attitudes at the peak of the pandemic as a spill-over effect [9,11], we were not able to proof a concurrent change in the change in influenza, pneumococci, and COVID-19 vaccination attitudes over time. On the contrary, in our study, risk groups showed a fairly stable, moderately positive attitude towards the influenza and pneumococcal vaccination, which, however, was not translated into a positive decision to get vaccinated in every individual. 

The observed associations between the influenza, pneumococcal and COVID-19 vaccination attitudes, on the other hand, are in line with the results from other studies [10] and persisted between the two survey time points. Furthermore, previous studies reported an increased risk perception for other infections at the subsiding pandemic, which was also detectable in our data. Specifically, only two aspects of risk perception (danger of the disease and risk to get an influenza or pneumococcal infection) were stable predictors over time of attitudes towards both influenza and pneumococcal vaccination in our survey. In line with this, a previous survey among college studies showed that the perceived risk to contract influenza in 2023 as a facilitator of vaccination intention was increased compared to previous years, which may be due to the non-existent influenza seasons in 2020/2021 and 2021/2022 [34]. 

Our results may have different implications for clinical practice: First, there appears to be a constant moderately positive vaccination attitude towards influenza and pneumococcal vaccination among high-risk patients. Given that vaccination uptake is nevertheless too low [35], our results support previous findings [36] that suggest that vaccination campaigns may benefit from a focus on vaccination knowledge and information on the susceptibility for and dangers of influenza and pneumococcal infections as important facilitators of vaccination attitude. Second, it may be synergistic to focus campaigns not on single vaccines but on COVID-19 and influenza or pneumococci, for example, together. This approach is supported by the results of a previous US study that found that messages that advocated for the influenza and COVID-19 vaccinations favourably influenced the intention towards the other vaccination [37]. This, however, may not imply focusing on the co-administration of vaccines, which is viewed sceptically by many patients [38].

This study has several strengths. First, the longitudinal design allows for analyses of change in vaccination attitudes over a time frame of two years of an evolving and subsiding pandemic. Second, the sample covers a broad spectrum of risk groups with different chronic diseases. These risk groups are formed by the most important recipients of vaccinations against infectious diseases against which permanent immunity cannot be achieved. Our study has also important limitations to consider. First, with our study starting in 2021, we lack data on the pre-pandemic vaccination attitudes and, therefore, cannot include the baseline levels in our trend analyses over the whole course of the COVID-19 pandemic. Second, we included a convenience sample of risk groups with a vaccination recommendation for COVID-19, influenza and pneumococci; thus, the generalization of the results to the German population may be limited. Furthermore, we faced a certain drop-out between the first and second surveys. The resulting missings could nevertheless be considered in the evaluation by using structural equation modelling with FIML as a state of the art method for the analysis of data with planned missing values by design. Third, we used data from an observational longitudinal study, which do not allow for causal interpretations of the observed relationships between the variables in our study. Fourth, since the same items were used to record the vaccination attitudes (influenza/pneumococci/COVID-19), we cannot rule out the possibility that the associations found between vaccination attitudes towards influenza and pneumococci, and COVID-19 vaccination attitudes are subject to a common-method bias. Fifthly, since some of the respondents were interviewed by telephone or in person and the topic of vaccination was the subject of controversial public debate at the time of the interviews, we cannot rule out response bias for reasons of social desirability.

## 5. Conclusions

Attitudes towards influenza and pneumococcal vaccinations are closely linked to the perceived danger of the disease and the perceived risk of infection. Even during the subsiding pandemic, vaccination attitudes towards COVID-19 was constantly associated with the influenza and pneumococcal vaccination attitude. Based on the results, it seems sensible to take these aspects into account when designing future vaccination campaigns for at-risk patients, e.g., by incorporating synergistic campaign messages on different pathogens.

## Figures and Tables

**Figure 1 vaccines-12-01080-f001:**
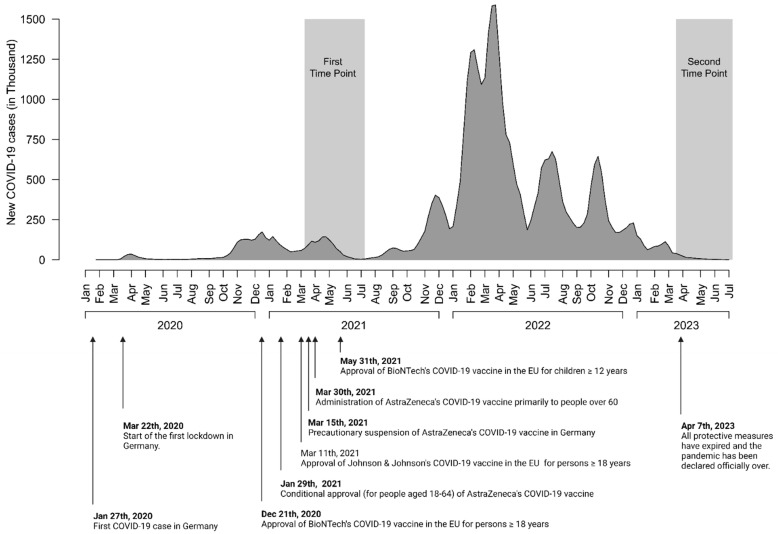
Timeline of the COVID-19 pandemic and survey time points.

**Table 1 vaccines-12-01080-t001:** Demographics of the survey population (N = 543).

Variables	Descriptive Statistics
Female gender, n (%)	276 (49.2)
Age in years, mean ± SD	52.6 ± 13.0
Education level, n (%)	
Low	111 (20.4)
Intermediate	223 (41.1)
High	209 (38.5)
Employment status, n (%)	
Unemployed	232 (42.7)
Employed	311 (57.3)
Health insurance, n (%)	
Private	64 (11.8)
Statutory	476 (87.7)
Not answered	3 (0.5)
Comorbidities	
Cancer, n (%)	157 (28.9)
Type 1 diabetes, n (%)	55 (10.1)
Type 2 diabetes, n (%)	74 (13.6)
Chronic heart failure, n (%)	67 (12.3)
Chronic bronchitis, n (%)	70 (12.9)
Chronic renal failure, n (%)	56 (10.3)
Chronic liver disease, n (%)	47 (8.7)
Chronic neurological disease, n (%)	1 (0.2)
Overweight, n (%)	19 (3.5)
Severe rheumatic disease, n (%)	52 (9.6)
Severe psoriasis, n (%)	50 (9.2)
HIV infection, n (%)	81 (14.9)
Asplenia, n (%)	0 (0.0)
Autoimmune disease, n (%)	2 (0.4)
Vaccination status influenza, n (%)	
Vaccinated (2021)	226 (41.6)
Vaccinated (2023) *	103 (39.6)
Vaccination status pneumococci, n (%)	
Vaccinated (2021)	87 (16.0)
Vaccinated (2023) *	46 (17.7)

* Sample size of the survey population at 2023 (N = 260).

**Table 2 vaccines-12-01080-t002:** Means of vaccination attitude, risk perception and vaccination knowledge for influenza and pneumococcal in 2021 and 2023.

		Influenza										Pneumococcal								
		2021			2023			Change				2021			2023			Change		
		M_1_	SE		M_2_	SE		M_2_ − M_1_	SE	*p*-Value		M_1_	SE		M_2_	SE		M_2_ − M_1_	SE	*p*-Value
Vaccination Attitude		3.384	0.043		3.382	0.049		−0.002	0.033	0.958		3.116	0.043		3.055	0.046		−0.061	0.036	0.094
Danger Disease		2.472	0.046		2.767	0.055		0.294	0.044	<0.001 *		2.671	0.058		2.716	0.058		0.045	0.050	0.374
Risk to Get		3.206	0.049		3.364	0.051		0.158	0.045	<0.001 *		2.495	0.050		2.687	0.056		0.193	0.048	<0.001 *
Danger Vaccination		1.952	0.038		2.015	0.038		0.063	0.034	0.064		2.040	0.039		2.030	0.038		−0.010	0.036	0.787
Effectiveness Vaccination		4.092	0.036		3.909	0.040		−0.183	0.037	<0.001 *		4.078	0.036		3.870	0.040		−0.208	0.037	<0.001 *
Vaccination Knowledge		0.757	0.021		0.773	0.026		0.016	0.025	0.516		0.467	0.024		0.514	0.028		0.047	0.022	0.035 *

* Significant on a significance level α = 0.05.

**Table 3 vaccines-12-01080-t003:** Standardized regression coefficients for predictors of influenza vaccination attitude in 2021 and 2023.

		2021				2023				Change		
		β_1_	SE	*p*-Value		β_2_	SE	*p*-Value		β_2_ − β_1_	SE	*p*-Value
Danger of disease		0.204	0.036	<0.001 *		0.164	0.057	0.004 *		−0.033	0.055	0.555
Risk to get disease		0.224	0.041	<0.001 *		0.337	0.061	<0.001 *		0.150	0.067	0.025 *
Danger of vaccination		−0.083	0.040	0.036 *		0.052	0.067	0.434		0.168	0.100	0.092
Effectiveness of vaccination		0.150	0.049	0.002 *		0.078	0.071	0.270		−0.076	0.108	0.479
Vaccination knowledge		0.125	0.036	<0.001 *		0.059	0.047	0.209		−0.160	0.137	0.244
COVID-19												
Danger of vaccination		0.011	0.042	0.792		−0.004	0.073	0.959		−0.016	0.094	0.866
Effectiveness of vaccination		−0.140	0.050	0.005 *		−0.095	0.063	0.130		0.065	0.090	0.471
Vaccination attitude		0.243	0.041	<0.001 *		0.232	0.063	<0.001 *		−0.034	0.071	0.629
Confidence		0.193	0.050	<0.001 *		0.088	0.057	0.122		−0.114	0.092	0.217
Calculation		−0.028	0.026	0.273		−0.022	0.042	0.589		0.002	0.059	0.978
Constraints		−0.053	0.032	0.092		−0.034	0.053	0.520		0.016	0.077	0.837
Complacency		−0.038	0.037	0.300		−0.102	0.055	0.066		−0.102	0.085	0.231
Collective responsibility		0.030	0.044	0.492		0.093	0.055	0.090		0.078	0.070	0.270

Change of the regression coefficients β_2_ − β_1_ is based on the unstandardized regression coefficients. * Significant on a significance level α = 0.05

**Table 4 vaccines-12-01080-t004:** Standardized regression coefficients for predictors of pneumococcal vaccination attitude in 2021 and 2023.

		2021				2023				Change		
		β_1_	SE	*p*-Value		β_2_	SE	*p*-Value		β_2_ − β_1_	SE	*p*-Value
Danger of disease		0.180	0.052	<0.001 *		0.218	0.069	0.002 *		0.051	0.061	0.407
Risk to get disease		0.210	0.050	<0.001 *		0.238	0.068	<0.001 *		0.030	0.063	0.639
Danger of vaccination		−0.068	0.052	0.190		−0.061	0.059	0.302		0.000	0.090	0.997
Effectiveness of vaccination		0.076	0.077	0.320		0.093	0.079	0.235		0.024	0.121	0.841
Vaccination knowledge		0.230	0.043	<0.001 *		0.077	0.05	0.124		−0.305	0.114	0.008 *
COVID-19												
Danger of vaccination		−0.012	0.057	0.839		0.112	0.067	0.092		0.126	0.089	0.156
Effectiveness of vaccination		−0.053	0.076	0.480		−0.050	0.071	0.480		0.011	0.109	0.916
Vaccination attitude		0.161	0.050	0.001 *		0.258	0.06	<0.001 *		0.064	0.069	0.355
Confidence		0.107	0.053	0.044 *		0.132	0.065	0.042		0.039	0.092	0.674
Calculation		−0.062	0.030	0.042 *		−0.038	0.042	0.360		0.022	0.052	0.670
Constraints		−0.109	0.035	0.002 *		−0.024	0.043	0.576		0.092	0.058	0.113
Complacency		−0.076	0.043	0.078		−0.039	0.048	0.424		0.045	0.075	0.552
Collective responsibility		0.010	0.051	0.838		0.076	0.060	0.207		0.070	0.075	0.355

Change of the regression coefficients β_2_ − β_1_ is based on the unstandardized regression coefficients. * Significant on a significance level α = 0.05

## Data Availability

The datasets used and/or analyzed during the current study are available from the corresponding author on reasonable request. Access to the anonymized data might be granted following review and permission by the ethics commission and data protection officer of the Jena University Hospital.

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
