# Peer review of "Changes in Attitudes towards Influenza and Pneumococcal Vaccination during the Subsiding COVID-19 Pandemic—Results of a Longitudinal Survey Study among Risk Groups in Germany between 2021 and 2023"

_vaccines, 2024, doi:10.3390/vaccines12091080_

Round 1

Reviewer 1 Report

Comments and Suggestions for Authors

The present manuscript is very important with clinical impact in the present era of the post pandemic COVID-19 and many emerging threats. The information about hesitancy of people at risk to vaccinate is very crucial in order to design campains.

The article is well written; however, the only flaw is the small number of individuals, especially during the second survey, including a small sample size of several risk groups for infection, and firm conclusions could not be extracted

Author Response

Comment 1: The present manuscript is very important with clinical impact in the present era of the post pandemic COVID-19 and many emerging threats. The information about hesitancy of people at risk to vaccinate is very crucial in order to design campaigns.

The article is well written; however, the only flaw is the small number of individuals, especially during the second survey, including a small sample size of several risk groups for infection, and firm conclusions could not be extracted

Response 1: We thank the reviewer for the remark. We agree that the sample size is a limitation of our study, especially in the second survey. For this reason, we opted for full information maximum likelihood estimation (FIML) as analysis method, which allows for inclusion of all participants including those with incomplete data due to non-participation at the second time point. We also made sure that, despite the reduced sample in the second survey, the proportion of patients with certain chronic diseases was the same across both surveys. To clarify this, we have revised the section “Sample recruitment and survey conduction”, which now reads:

“For the first wave of the longitudinal study, the market research institute IPSOS recruited a sample of 543 persons from Germany aged at least 18 years. In accordance with the study design, 260 of the initial 543 persons were surveyed again in the third wave. The recruitment process was designed to reach a predetermined proportion of patients with certain chronic diseases (e.g. diabetes, chronic diseases, cancer) in each wave. The proportions were the same across all waves.”

Furthermore, the limited sample size was also added as a limitation in the discussion:

“Furthermore, we faced a certain drop-out between the first and second survey. The resulting missings could nevertheless be taken into account in the evaluation by using structural equation modelling with FIML as a state of the art method for the analysis of data with planned missing values by design.”

Reviewer 2 Report

Comments and Suggestions for Authors

Thank you very much for the opportunity to review this well prepared and written manuscript. Born et al. studied change in attitude towards influenza and pneumococcal vaccination during COVID-19. The study is a longitudinal survey with a large sample size, which give the study significant credibility. The study is well-structured, meticulously conducted, and provides insightful data that could have significant implications for both clinical practice and future research. 

The main point (minor) I have is with the survey development. The author did explain some details about the development of the survey, however, it is not clear how they formulate the questions. How do they know that these questions gave them the correct description of people attitude, which is not easy to capture? They do not provide any reference for the survey, nor have they validated their survey for reliability and validity. This is very important. Authors need to show that the questions do address the attitude of the participants. Also, they need to use some statistics such as Cronbach’s alpha (for scaled questions).

Discussion: Well written, but it has very little comparisons with other findings from international studies. 

Limitations: Please expand on the limitation of the instrument you used (survey) such as response bias and the accuracy of self-reporting. 

Author Response

Thank you very much for the opportunity to review this well prepared and written manuscript. Born et al. studied change in attitude towards influenza and pneumococcal vaccination during COVID-19. The study is a longitudinal survey with a large sample size, which give the study significant credibility. The study is well structured, meticulously conducted, and provides insightful data that could have significant implications for both clinical practice and future research.

Comment 1: The main point (minor) I have is with the survey development. The author did explain some details about the development of the survey, however, it is not clear how they formulate the questions. How do they know that these questions gave them the correct description of people attitude, which is not easy to capture? They do not provide any reference for the survey, nor have they validated their survey for reliability and validity. This is very important. Authors need to show that the questions do address the attitude of the participants. Also, they need to use some statistics such as Cronbach’s alpha (for scaled questions).

Response 1: Thank you for the notion. In order to clarify the content and wording of the questions, we have attached the questionnaire as a supplement. All questions and scales used in the present study were already used in previous studies. We added a sentence in the section “Survey development” to clarify this point and listed the corresponding references. Furthermore, we calculated McDonald’s Omega for the scaled questions, because it is superior in cases of unequal factor loadings and missing item responses. Especially for the vaccination attitude scales, the reliability was very high.

This was added to the results section:

“For the vaccination attitude scales, McDonald's Omega reliability estimate ranged from 0.91 to 0.96, which indicates a high reliability. For more details see Table S1.”

Comment 2: Discussion: Well written, but it has very little comparisons with other findings from international studies.

Response 2: Thank you for this feedback. We revised the discussion to clarify and added further references.

Kwetkat, A., et al., [Current recommendations for vaccination in older adults]. MMW Fortschr Med, 2021. 163(10): p. 42-49.

Achterbergh, R.C.A., I. McGovern, and M. Haag, Co-Administration of Influenza and COVID-19 Vaccines: Policy Review and Vaccination Coverage Trends in the European Union, UK, US, and Canada between 2019 and 2023. Vaccines, 2024. 12(2): p. 216.

Comment 3: Limitations: Please expand on the limitation of the instrument you used (survey) such as response bias and the accuracy of self-reporting.

Response 3: Thanks for the remark. We have added this point to the discussion: “Fifth, as some of the respondents were interviewed by telephone or in person and the topic of vaccination was controversial discussed in the society at the time of the interviews, we cannot rule out a response bias due to social desirability.”

Reviewer 3 Report

Comments and Suggestions for Authors

This is a good idea with adequate material, that resulted in a good paper. I read easily, the methods are well explained, and the results are well explored. Discussion was focused on explaining reasons for adherence to respiratory disease vaccination

Author Response

Comment 1: This is a good idea with adequate material that resulted in a good paper. I read easily, the methods are well explained, and the results are well explored. Discussion was focused on explaining reasons for adherence to respiratory disease vaccination

Response 1: We thank the reviewer for the feedback.

Reviewer 4 Report

Comments and Suggestions for Authors

My question what were the status of participants on the two vaccines?

Did you ask if they were infected by COVD-19 and pneumococcal infections?

Did the questions from the survey ask the participants if they personally knew of someone who had the diseases?

Author Response

Comment 1: My question what were the status of participants on the two vaccines?

Response 1: Thanks for the remark. We added the immune status regarding influenza and pneumococci for 2021 and 2023 in Table 1 and reported the temporal trends in the results and discussion.

 “The vaccination rate in our survey population for influenza in 2021 (41.6%) was significantly higher than the vaccination rate for pneumococcal vaccination (16.0%). In 2023 the vaccination rates for influenza and pneumococcal vaccination remained almost unchanged in 2023 (39.6% and 17.7%).”

Comment 2: Did you ask if they were infected by COVD-19 and pneumococcal infections?

Response 2: Unfortunately, we do not record any information regarding a COVID-19 or pneumococcal infection status in the two surveys.

Comment 3: Did the questions from the survey ask the participants if they personally knew of someone who had the diseases?

Response 3: We agree this is an important information, however, such question was not part of our survey.

Round 2

Reviewer 4 Report

Comments and Suggestions for Authors

You addressed most of my concerns